# Artificial Intelligence-Based Optimal Grasping Control

**DOI:** 10.3390/s20216390

**Published:** 2020-11-09

**Authors:** Dongeon Kim, Jonghak Lee, Wan-Young Chung, Jangmyung Lee

**Affiliations:** 1Department of Electronics Engineering, Pusan National University, Jangjeon-dong, Geumjeong-gu, Busan 609-735, Korea; dongeon1696@pusan.ac.kr (D.K.); jonghak1696@pusan.ac.kr (J.L.); 2Department of Electronics Engineering, Pukyong National University, Daeyeon 3-dong, Nam-gu, Busan 608-737, Korea; wychung@pknu.ac.kr

**Keywords:** tactile sensing module, deep neural network, air pressure sensor

## Abstract

A new tactile sensing module was proposed to sense the contact force and location of an object on a robot hand, which was attached on the robot finger. Three air pressure sensors are installed at the tip of the finger to detect the contacting force at the points. To obtain a nominal contact force at the finger from data from the three air pressure sensors, a force estimation was developed based upon the learning of a deep neural network. The data from the three air pressure sensors were utilized as inputs to estimate the contact force at the finger. In the tactile module, the arrival time of the air pressure sensor data has been utilized to recognize the contact point of the robot finger against an object. Using the three air pressure sensors and arrival time, the finger location can be divided into 3 × 3 block locations. The resolution of the contact point recognition was improved to 6 × 4 block locations on the finger using an artificial neural network. The accuracy and effectiveness of the tactile module were verified using real grasping experiments. With this stable grasping, an optimal grasping force was estimated empirically with fuzzy rules for a given object.

## 1. Introduction

Among the pieces of information that are received as feedback during robot movements, vision and tactile sensations are the most useful. When there is information on direct contact with a target object, both the target object and robot can be protected, and the operation can be performed safely. Tactile sensing is mainly used to judge the hardness of objects that are touched or to sense the temperature to carry out additional avoidance actions. Studies have been investigating tactile sensors for humanoid robot hands for mimicking human tactile sensing [1,2,3,4,5]. In particular, methods have been proposed for collecting tactile sensing information from various sensors fused together and for developing a tactile sensing module for diverse body parts [6,7]. However, as tactile sensors are still in the development stage and the manufacturing costs of tactile sensors are high in many cases, it is difficult to find commercialized products using tactile sensors [8,9]. Moreover, there are tactile sensors for robots that function based on pressure sensors actuated by fluids [10,11].

There is also a method of configuring a tactile sensor as a force sensing resistor (FSR) [12,13]. A method using a super resolution algorithm has also been proposed, although a lower spatial resolution is measured compared with other types of force or pressure sensors [14]. However, both methods are not sufficiently portable to be applied to other systems.

Unlike existing tactile sensors, a tactile sensing module constructed with air pressure sensors (as developed by us) has a similar response speed and wider sensing range than an FSR [15]. In addition, it can be manufactured at a cost several dozen times lower than that of tactile sensors. It can also perform diverse applied operations according to the temperature of an object through the temperature sensing function and can measure tactile sensations more precisely.

In robot gripping operations, neural networks are being universalized. In most research, vision is used as the input value of the system [16,17]. In particular, the research that used an optical-based tactile sensor called GelSight is impressive [18]. Our system is robust against the effects of object characteristics or the environment. It does not require light or sound-related characteristics when measuring the pressure value in contact with a robot.

This paper is organized as follows. Section 2 discusses the sensing of contact forces using pressure sensors. In Section 3, an arrival of time (AoT) algorithm is proposed for sensing the degree of contact of objects that are touched by the robot finger through the developed tactile sensor. In Section 4, the resolution is improved by configuring a neural network, and the improvement is indicated through a contact map. In Section 5, the object is grasped with the optimum grasping force based on the fuzzy controller, after detecting the contact force. Section 7 discusses performing adaptive grasping for different objects using the developed tactile sensor.

## 2. Sensing of Contact Force through Air Pressure Sensors

### 2.1. Configuration of Tactile Sensing Module through Air Pressure Sensors

In this study, to sense touches with a gripper-type robot hand composed of three fingers, a tactile sensing module is developed for implantation at the tip of a finger, as shown in Figure 1. The developed tactile sensor can be implanted in parts corresponding to the thumb and the middle finger. Considering the structure in which the robot finger skin is implanted, a structure made of silicone was designed; three MS5611 sensors can be inserted therein. Using the aforementioned method, the degree to which the robot hand touches an object can be sensed [19].

As shown in Figure 2, the skin of the robot finger was configured with a silicone-based material, using a material capable of (and suitable for) transmitting external shocks, while withstanding said shocks. In addition, to transmit shock waves well with a softer material, the inside was made using liquid silicone and solid silicone in a 1:1 ratio; thus, the material was similar to a solid jelly material, thereby facilitating the sensing of touches by the air pressure sensors.

When grasping an object that is a target to be worked on, the weight of the object should be sensed and an appropriate force should be applied to perform stable grasping. To this end, initially, the weight of the touched object is sensed through the air pressure sensor. Thereafter, the output values of the air pressure sensor according to the weights are learned through a deep neural network (DNN). For the experimental conditions, the mass is constantly increased, but the temperature and surface area of the object remain the same. Based on the aforementioned information, the linearized weights can be predicted, according to the outputs of the sensor. The experiment was conducted by continuously adding and replacing 10 g weights in a cylindrical storage box, as shown in Figure 3.

### 2.2. Neural Network Configuration for Predicting Contact Force

When the weight of an object is linearly increased and measured using a module composed of air pressure sensors, sections may exist where the values stand out non-linearly. For the module to function as a sensor, such nonlinearity should be linearized. This is achieved by configuring a DNN as shown in Figure 4, and the values for the weights of the object are predicted through learning.

The object function for training the DNN is defined as the mean squared error of the neural network output of the training data and the actual object weight. The structure is as follows:(1)$$\begin{array}{ll} MSE(\theta ,b) & =\frac{1}{m}\sum_{i=1}^{m}{\left({\hat{y}}_{i}-y_{i}\right)}^{2}\\ & =\frac{1}{m}\sum_{i=1}^{m}{\left(\theta \cdot x_{i}+b-y_{i}\right)}^{2} \end{array}$$
where *m* represents the number of training data sets, ${\hat{y}}_{i}$ represents the *i*th predicted value, $y_{i}$ represents the *i*th label, $x_{i}$ represents the *i*th input data vector, θ represents the weight vector, and *b* represents the bias.

The DNN consists of an input layer, hidden layer, and output layer. The variables of the three air pressure sensors are the input elements; the linear weight gain values are input as $y_{i}$ for supervised learning. For the output from one layer to the next, the input values from the preceding layer are multiplied by the weight (i.e., the connection strength), b is added to obtain the element values, and the activation function is applied to obtain the parameter value of the next layer to obtain the value of θ. Here, sigmoid is used as the activation function for the input and output layers, and a rectified linear unit (ReLU) and sigmoid are used for the hidden layer. Adam is used as the optimizer [20]. The weights are updated through the gradient descent method, as shown in Equation (2). The gradient descent method updates the weights through a product of the differential value of the objective function and learning rate and modifies the weights in all hidden layers to reduce the error(s) output from the final layer.
(2)$$\omega =\omega -\eta \frac{\partial }{\partial \omega }MSE(\theta ,b)$$
where ω is a weight vector and η is a learning rate (used when applying the current variation of ω to the update of ω). The proposed DNN performs repetitive learning using a backpropagation algorithm.

Table 1 presents the weights of the object as predicted when the average measured values of the air pressure sensors that change according to the linear weight increase are linearly distributed, and the relevant values are input as described above. Accordingly, linear output values are obtained; thus, the sensors can function as sensors.

## 3. Touch Sensing Using Arrival of Time (AoT) Algorithm

When robot fingers grasp an object, the contact position may vary depending on the shape of the object, position where the object is placed, and posture of the fingers. The AoT algorithm is based on using the position and characteristics of the air pressure sensor constituting the touch sensing unit. Using the algorithm, the touch sensing unit can sense the position where the object is touched, in addition to predicting the weight of the object [21]. The AoT algorithm senses a touch position basis the sensing time of each sensor and distance value (r) between each sensor, as shown in Figure 5. In this study, the touch sensing unit was divided into nine equal-sized rectangles, and the positions of the air pressure sensors were produced (as shown by the P points) in an inverted triangular shape, in consideration of the binding structure with the robot hand frame. The divided area was denoted as $Arr_{n}$, and the distances between the center point of each area and sensors were denoted by $r_{n,k}$.

The AoT algorithm for detecting an object when the object comes into contact with a touch sensing module is expressed as shown in Equation (3). It is calculated by multiplying the value obtained by dividing the measured air pressure value by $r_{n,k}$ with the total time during which the air pressure sensors sensed (as divided by sensor).
(3)$$Arr_{n}=\sum_{k=1}^{3}\left(\frac{P_{k}}{r_{n,k}}\cdot \frac{\sum_{x=1}^{3}t_{x}}{t_{k}}\right)$$
where $P_{k}$ refers to an air pressure value as measured by the *k*th air pressure sensor, $t_{x}$ refers to the sensing time of the air pressure sensor, and $t_{k}$ refers to the time sensed by the *k*th air pressure sensor after contact.

## 4. Enhancement of Sensing Resolution through Learning

To enhance the sensing precision of the contact area (as sensed according to the contact position and weight of the object), an artificial neural network is configured to learn the contact area. Through the foregoing, the resolution of the touch sensing can be increased from $Arr_{9}$ to $Arr_{24}$. The hidden layers of the artificial neural network are composed of 64 and 32 layers, respectively. ReLU is used as the activation function, and Adam is used as the optimizer. For the contact area input into the artificial neural network, as shown in Figure 6, the contact area $Arr_{n}$ (from the AoT algorithm parameters) is used. The existing $Arr_{9}$ with 3 × 3 resolution is learned sequentially in a clockwise direction according to the center of gravity, as shown in Figure 7. The learning is conducted again after increasing the resolution to 4 × 6.

Cases are further divided to build the training data by considering the contact area of the tactile sensing module with the contact object. The divided contact areas are first moved to the right by one block and are then moved to the bottom by one block when completed, i.e., they are learned sequentially, as shown in Figure 7. To change the contact weight, the contact part on the gripper is accurately sized to modify the contact area, as shown in Figure 8. To change the force exerted on each area, a precise-control gripper is used. Figure 9 shows that the experiment was conducted by varying the contact force to 5, 7, and 9 kg, among others, by changing the torque of the gripper. The size measured in Figure 9 corresponds to that in Figure 7e. The gripper used is the RH-P12-RN model from ROBOTIS.

In Figure 7, the contact areas of b and f are of the same size as that in Figure 7, but they are different cases. Hence, a unique number is assigned to each range, such as $Arr_{1}=1.1$, $Arr_{6}=1.6$, and $Arr_{24}=4.6$. A numerator is set by sequentially adding $Arr_{i}$, which is constructed depending on each contact shape, and the contact size is set as the denominator. The average contact area is calculated as shown in Equation (4).
(4)$$Average\_area=\sum_{i=1}^{n}\frac{Arr_{i}}{n}$$
where *n* refers to the size, and *i* refers to the unique number of $Arr_{i}$, which is the area unit depending on each contact shape.

Table 2 shows some of the training data for the average of the contact area defined in Equation (4).

In total, 10,000 training data are measured for each size, and the value is averaged through several experiments. As there are many similar values, only typical values are described as examples. Each pressure sensor has an initial value of 1010.xx. The decimals change randomly, but the integers have meaningful values depending on the contact force. The value may continue to increase during the experiment, and when the experiment is paused for approximately 2 min to correct this, the bias value is adjusted again to set the initial value to 1010.xx. In area learning, the area and position measured by each sensor can be inferred through an average value of the area, which is the last column of the table corresponding to the label value. To prevent the model’s overfitting, the L2 regularization was set to 0.001 [22], and the dropout ratio was set to 30%. Figure 10 shows the predicted result of the contact location through training. Since it is a model that has various parameters as input values and predicts the contacted location, it is suitable for MLP (Multi-Layer Perceptron) to use.

Figure 11 is divided into 24 areas of the touch sensing module, and the touches are expressed differently, according to the contact position with the object. When a touch is detected according to a contact position, it is expressed with a high concentration of red.

## 5. Fuzzy Controller for Optimal Grasping

The control system for controlling grasping is shown in Figure 12. This controller controls the robot hand using the optimum torque operating value when the pressure value output from the tactile sensing module located at the fingertip of the robot hand is grasped through the fuzzy controller [23].

In the developed touch sensing module, the difference between the average of the contact pressure values, the output value at the time of grasping, and its differential value are entered into the fuzzy controller. In Figure 13, the optimum grasping position and torque force are derived by using the outputs of the grasping torques. That is, to control the grasping force with the given output torque values, position control is implemented.

Figure 14 presents a fuzzy control system. The knowledge base refers to knowledge on the average input value of the pressure value and the amount of its change arising from contact, which is the input value, and knowledge on the membership function (MF) of the torque value during grasping, which is the output value. The rule base is information on the control rules of the robot hand. Fuzzification first makes a control decision by normalizing a crisp input value to a value between 0 and 1. The fuzzy inference engine infers the fuzzy output through the Mamdani method for an input value using the rules from the rule base and the membership functions of the database. Defuzzification generates a numerical value for the extent of torque during grasping by converting the fuzzy output from the fuzzy inference engine into a crisp value through the center average method. As a result of this, grasping position control is performed.

A membership function (MF) is constructed to calculate the goodness of fit of the fuzzy ranges, according to the contact pressure values. It is divided into 9 sections (NH, NB, NM, NS, ZO, PS PM, PB, PH); the meanings thereof are shown in Table 3.

Two types of MF are used for the average value of the pressure sensor, which is an input value, and the derivative of the average value. For NH and PH in Table 3, the Gaussian function is used as shown in Equation (5).
(5)$$f_{i}\left(x;\;\sigma_{i},\;\mu_{i}\right)=\;e^{\frac{-{\left(x-\mu_{i}\right)}^{2}}{2\sigma_{i}^{2}}}$$

Hence, among the seven MFs, the cases in which *i* is one and nine correspond to Equation (5). Here, $\sigma_{i}$ and $\mu_{i}$ refer to the standard deviation and average, respectively. The remaining MFs are defined as a trigonometric function, as shown in Equation (6).
(6)$$f_{i}\left(x;\;a_{i},\;b_{i},\;c_{i}\right)=\;\{\begin{array}{cc} 0, & x\leq a_{i}\;\\ \frac{x-a_{i}}{b_{i}-a_{i}}, & a_{i}\leq x\leq b_{i}\;\\ \frac{c_{i}-x}{c_{i}-b_{i}}, & b_{i}\leq x\leq c_{i}\\ 0, & c_{i}\leq x \end{array}$$
where *i* is 2, 3, …, 8; *b_i_* denotes the peak point of an MF; and $a_{i}$ and $c_{i}$ are the boundary values of the MF. The Gaussian function is used for the MF of the output value, as shown in Equation (5).

The composition of the entire fuzzy MF for the contact pressure value of the robot hand grasping, rate of change of the contact pressure value, and output force are as shown in Figure 15, Figure 16, Figure 17 and Figure 18.

## 6. Robot Hand Control System

The system is controlled by a PC and three microcontroller units (MCUs), as shown in Figure 19. The PC performs kinematics operation through a MATLAB graphic user interface and forwards the target position derived through inverse kinematics to an MCU. The MCU proceeds to control through the transmitted information and performs a feedback of the current information to the PC. The robot used herein has a gripper-type robot hand attached to a five-degree-of-freedom manipulator comprising five DC motors.

The specifications of the DC motors and the robot hand that comprises joints 1–5 of the robotic arm are shown in Table 4.

To control the robot hand equipped with the tactile sensing module shown in Figure 1, a control module is constructed, as shown in Figure 20. It is made up of two layers. The upper layer has an operation unit and a sensing unit, and the lower layer has a communication unit and a power supply unit.

On the upper layer, one MCU and a pressure sensor module for sensing a contact can be installed. One tactile sensing module requires a total of three pressure sensor modules, and a total of six pressure sensor modules can be mounted on one robot hand control module. This structure enables the control of the modules mounted on the thumb and middle finger simultaneously.

The control commands and encoder data between the MCU and the robot hand are converted through a communication converter between the serial communication interface (SCI) and RS 485. The pressure sensor at the fingertip transmits the sensed data to the MCU through the inter-integrated circuit (I2C) communication, and the MCU transmits the data to the PC through the controller area network (CAN) communication. The robot hand is controlled through this process. The robot hand uses the RS 485 communication, and the MCU uses the SCI, CAN, and I2C communications. As CAN communication is used when the manipulator is controlled through the PC, there is a communication unit, which includes a transceiver for level conversion when transmitting the data to control the robot hand. In addition, a DC–DC converter exists on the bottom layer of the module, as shown in Figure 21, for stable voltage supply to the pressure sensor, MCU, and the communication terminal.

## 7. Adaptive Grasping Experiment

A grasping experiment was conducted, based on a robot hand comprising a motor driven by a cortex m3. To configure diverse contact forces and areas, objects with diverse shapes and hardness levels were configured as grasping objects, as shown in Figure 22. When tissue paper (with a low level of hardness) or paper cups (with low strength) were grasped, the touch sensing value and current amount measured by the touch sensing module were measured to perform adaptive grasping through torque control. The maximum torque output limit of the motor for moving the robot fingers was set to 0 (0%)–1023 (100%). When grasping was conducted after setting the torque to 100%, the current continued to increase so that the grasping action was released, leading to a failure of grasping.

Figure 23 shows that the current value increases when a roll of tissue is grasped. The current value generated during the grasping depends on the object being grasped and degree of touch sensing; these are received as feedback to conduct grasping compliant with the set torque value.

Figure 24 shows the output values of the grasping torque and grasping position, as adjusted according to the contact force of the touch sensing module fed back when grasping the paper cup. Thus, it can be seen that optimum control of the grasping torque and position is conducted.

Table 5 shows the torque values set when the objects shown in Figure 21 above are grasped. The “torque min” is the value when the object is grasped so that it is not damaged, and the “torque max” is the value when the grasping release phenomenon does not occur, even if the grasping action continues for more than 2 min. The relevant values are set through feedback from the touch sensing module, and thus, adaptive grasping is conducted.

## 8. Discussion/Conclusions

A method for improving the sensitivity of a contact module equipped with a touch sensing function was proposed and implemented. The output values of the touch sensor (according to the weight values of the object) were linearized through DNN learning, and additional learning based on the contact areas and force enabled high-resolution touch sensing. The values of the proposed touch sensing module were received through feedback to conduct adaptive grasping operations, thereby showing the effectiveness of the touch sensor. In the future, optimal grasping torque control will be conducted for a more diverse set of general objects.

## Figures and Tables

**Figure 1 sensors-20-06390-f001:**
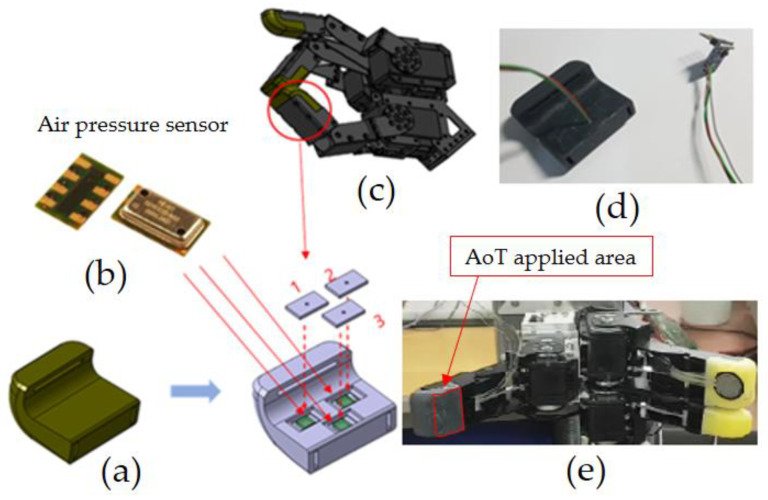
Configuration of tactile sensing module: (**a**) silicone base of the module, (**b**) air pressure sensor, (**c**) robot hand, (**d**) tactile sensing module, (**e**) module applied to robot hand.

**Figure 2 sensors-20-06390-f002:**
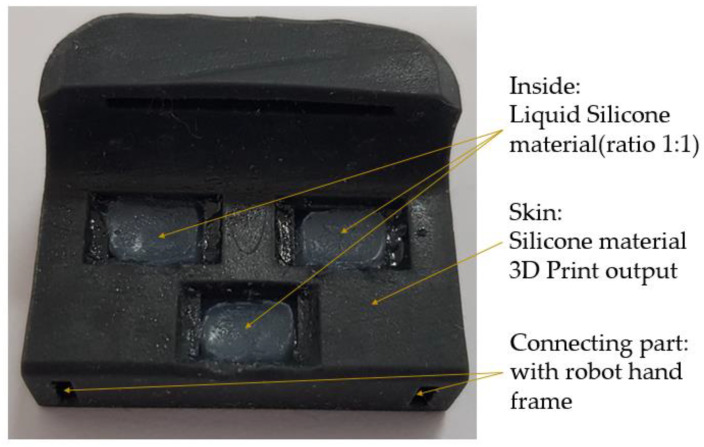
Configuration of finger skin.

**Figure 3 sensors-20-06390-f003:**
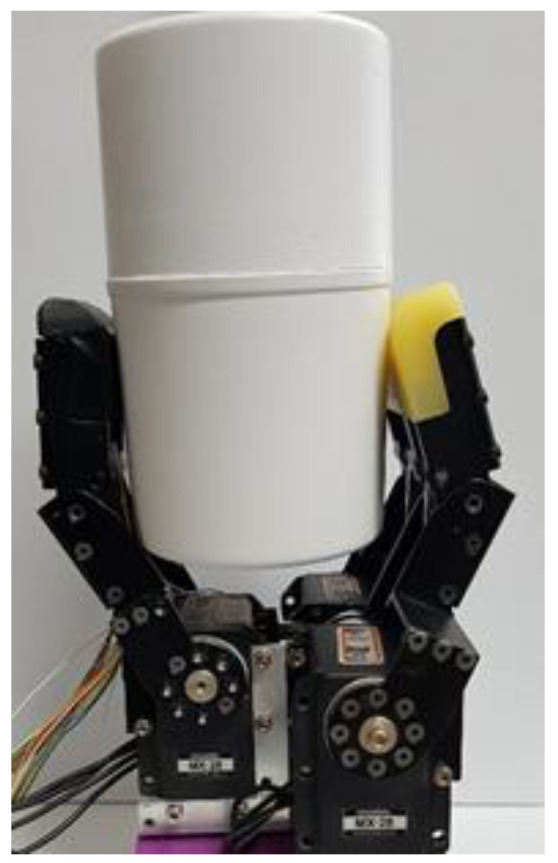
Object weight detection experiment.

**Figure 4 sensors-20-06390-f004:**
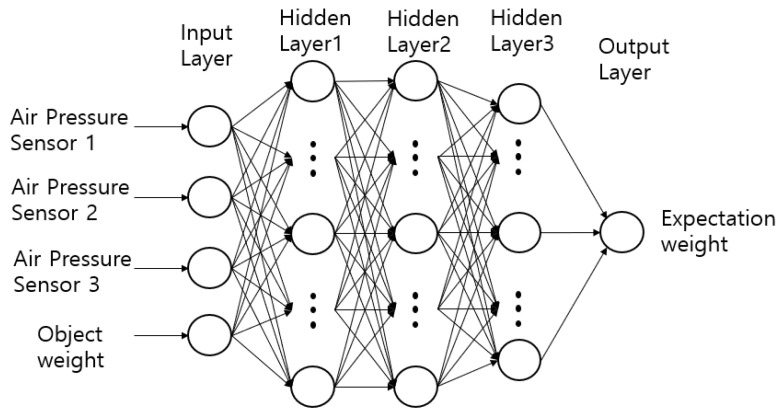
Weight sensing training with deep neural network.

**Figure 5 sensors-20-06390-f005:**
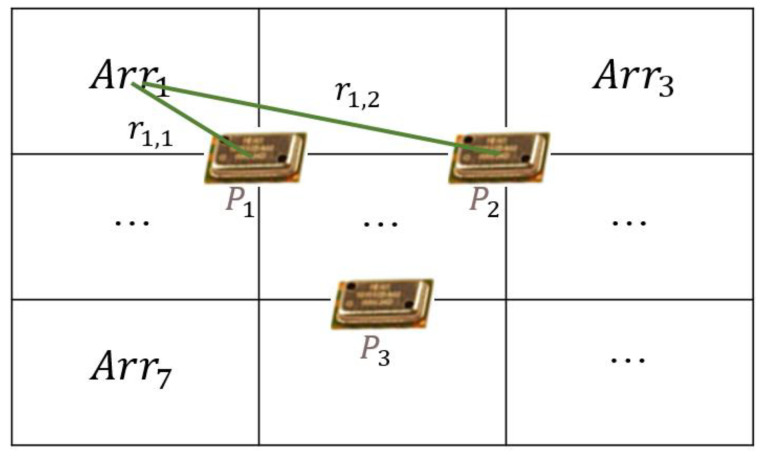
Parameters of arrival of time (AoT) algorithm.

**Figure 6 sensors-20-06390-f006:**
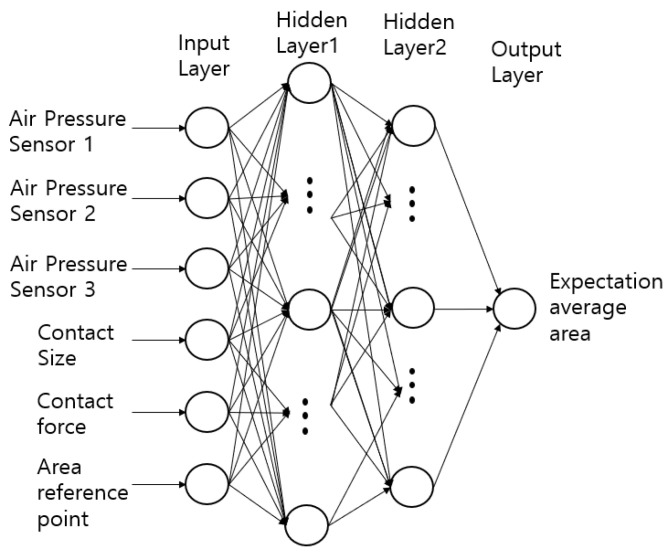
Artificial neural network for contact area training.

**Figure 7 sensors-20-06390-f007:**
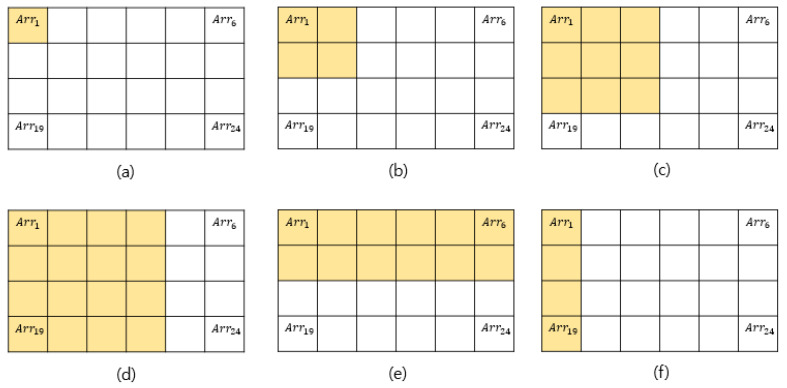
Range setting according to contact area: (**a**) size 1, (**b**) size 4–1, (**c**) size 9, (**d**) size 16, (**e**) size 12, (**f**) size 4–2.

**Figure 8 sensors-20-06390-f008:**
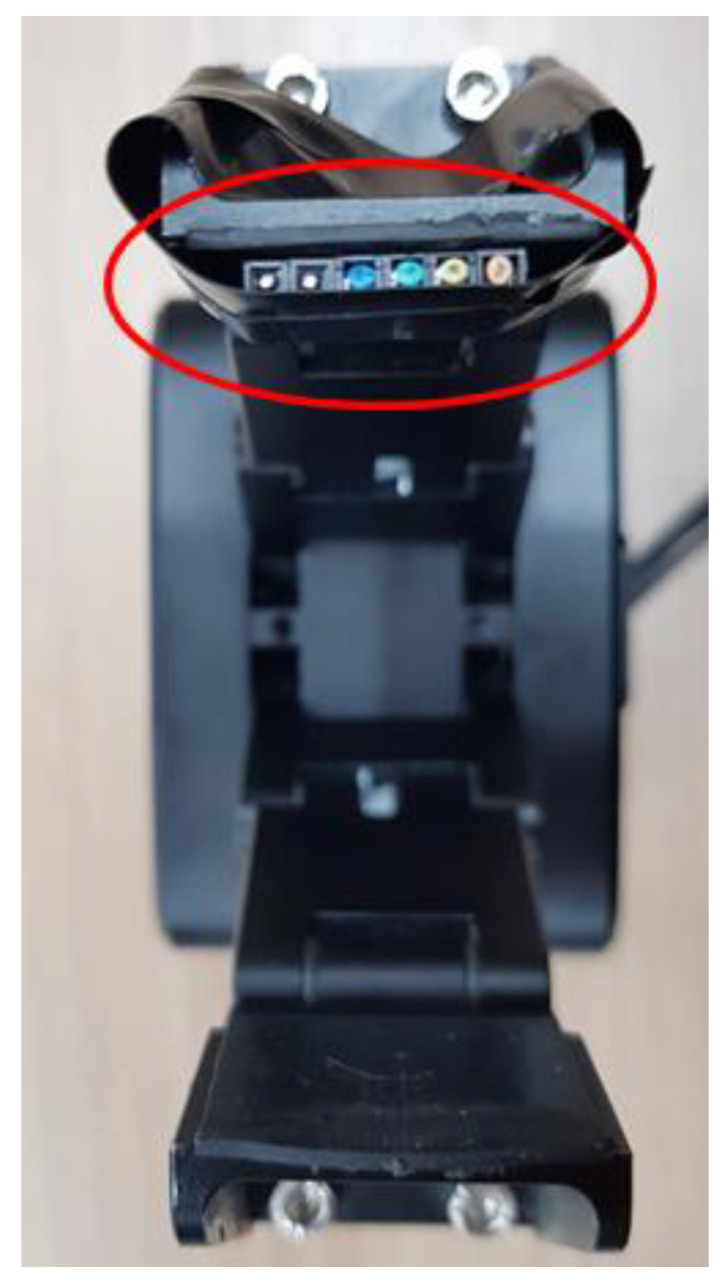
Structure for conversion of contact.

**Figure 9 sensors-20-06390-f009:**
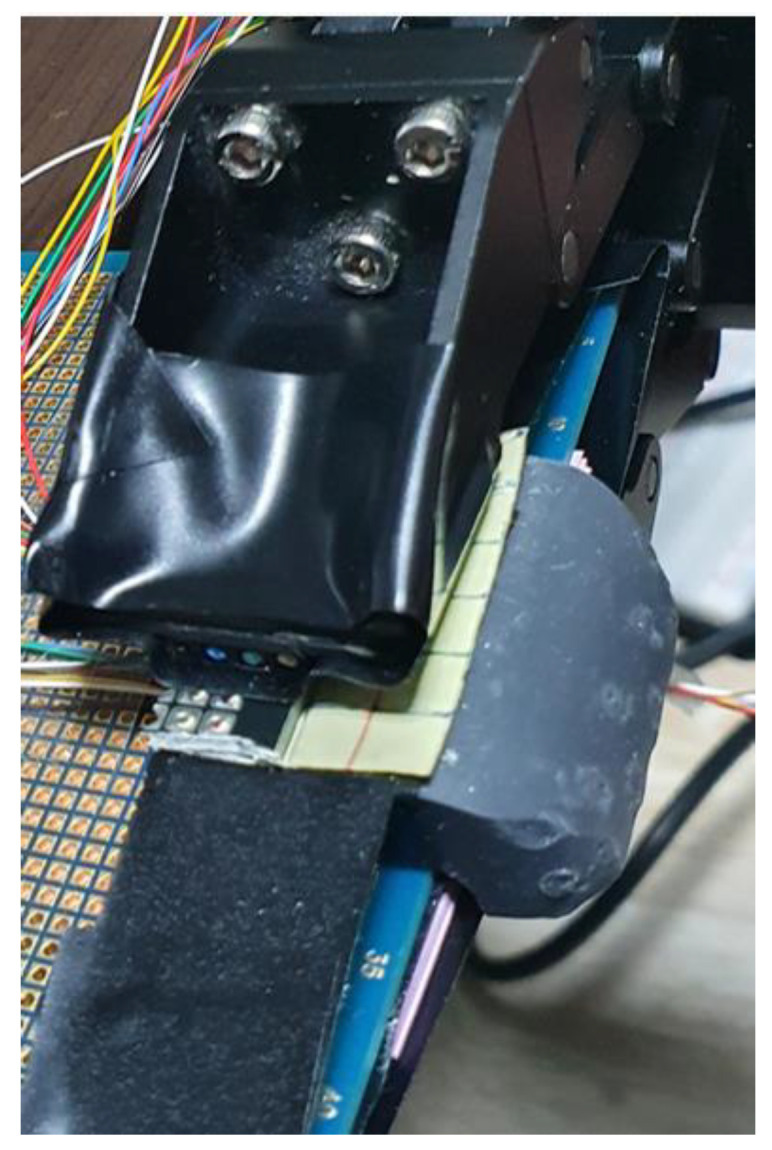
Experiment measured by converting contact area and force.

**Figure 10 sensors-20-06390-f010:**
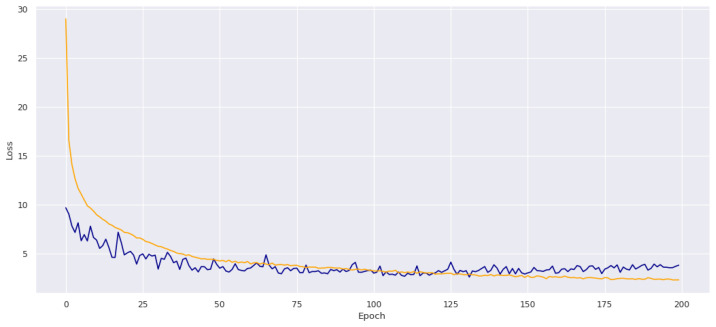
Results of contact area prediction through MLP training.

**Figure 11 sensors-20-06390-f011:**
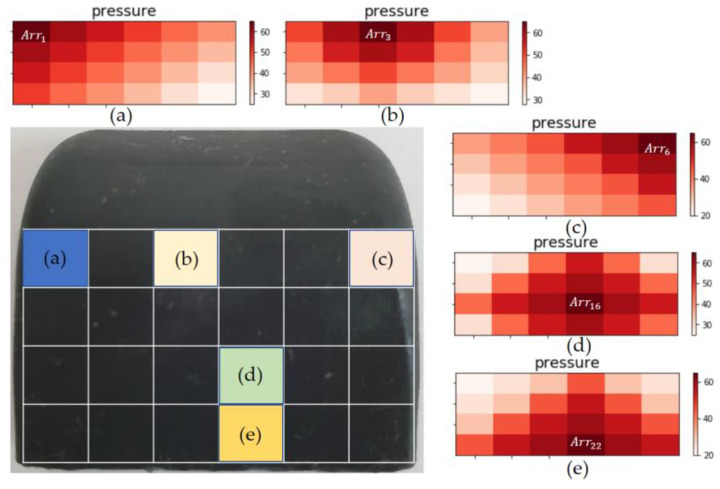
Sensing expression according to touch point of tactile sensing module.

**Figure 12 sensors-20-06390-f012:**
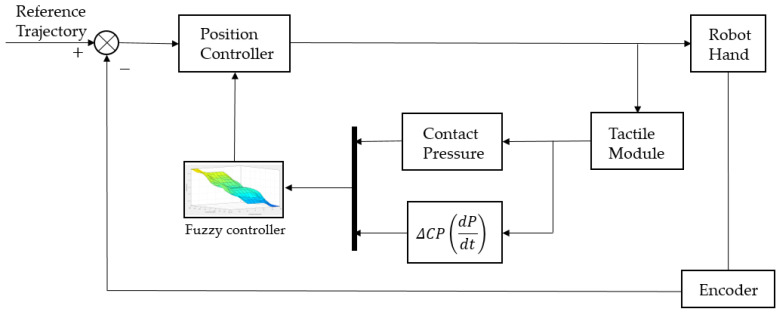
Optimal grasping controller.

**Figure 13 sensors-20-06390-f013:**
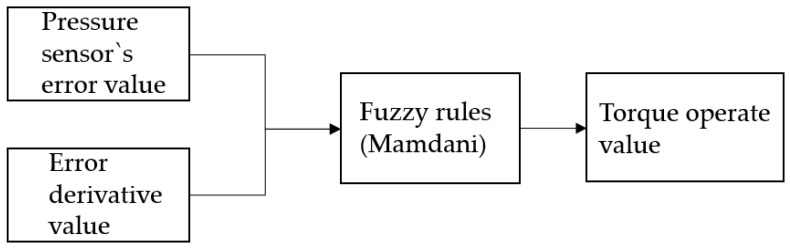
Fuzzy control system.

**Figure 14 sensors-20-06390-f014:**
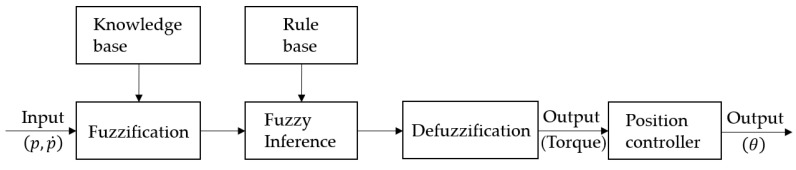
Fuzzy control system.

**Figure 15 sensors-20-06390-f015:**
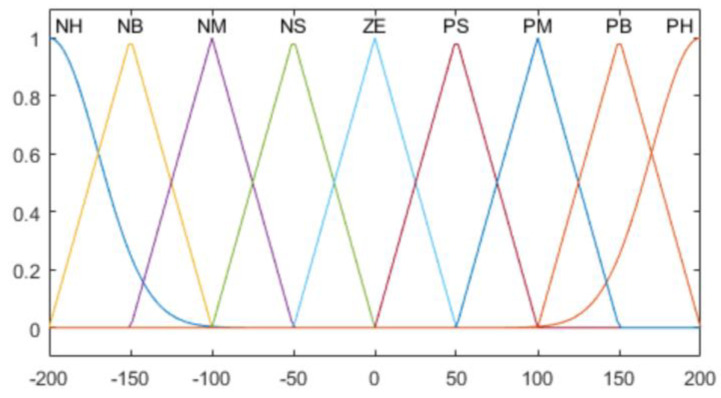
Contact pressure error membership functions (MFs).

**Figure 16 sensors-20-06390-f016:**
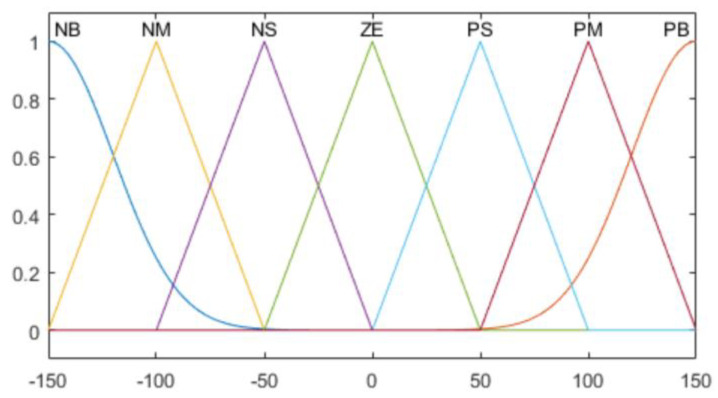
Contact pressure derivative MFs.

**Figure 17 sensors-20-06390-f017:**
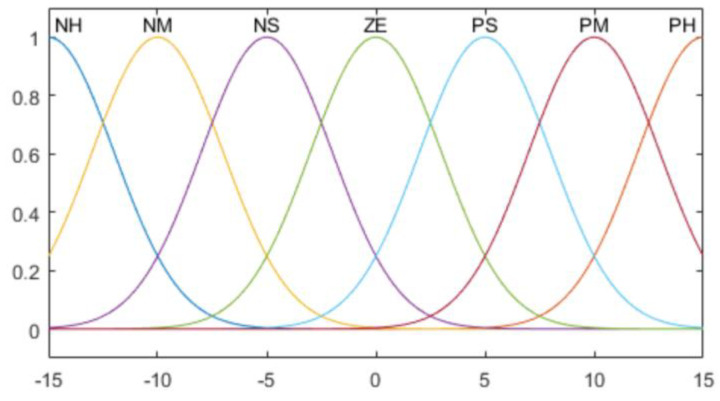
Optimal torque MFs.

**Figure 18 sensors-20-06390-f018:**
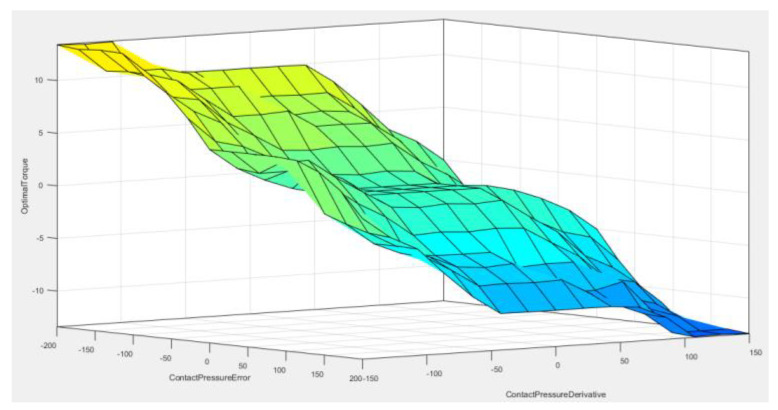
Surface of fuzzy controller.

**Figure 19 sensors-20-06390-f019:**
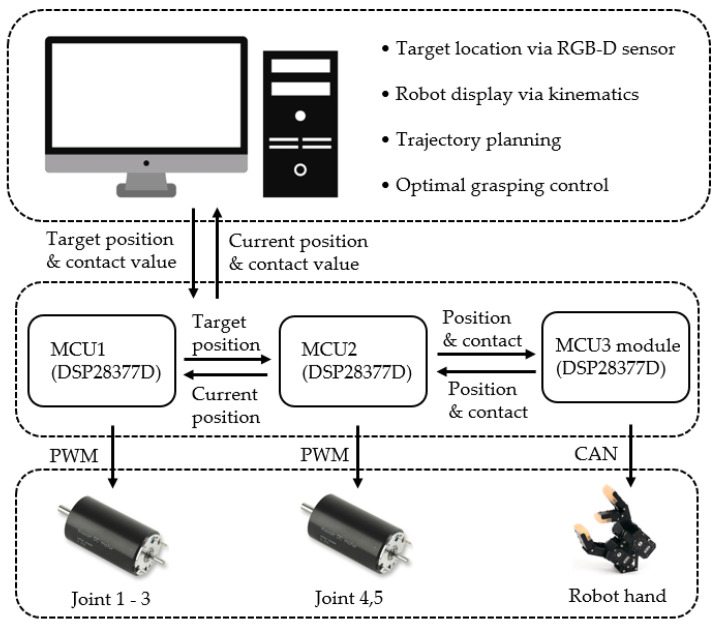
Control system configuration.

**Figure 20 sensors-20-06390-f020:**
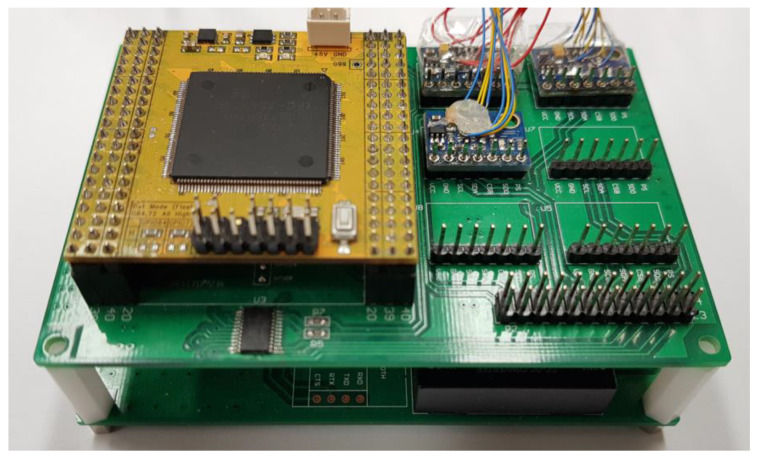
Robot hand control module.

**Figure 21 sensors-20-06390-f021:**
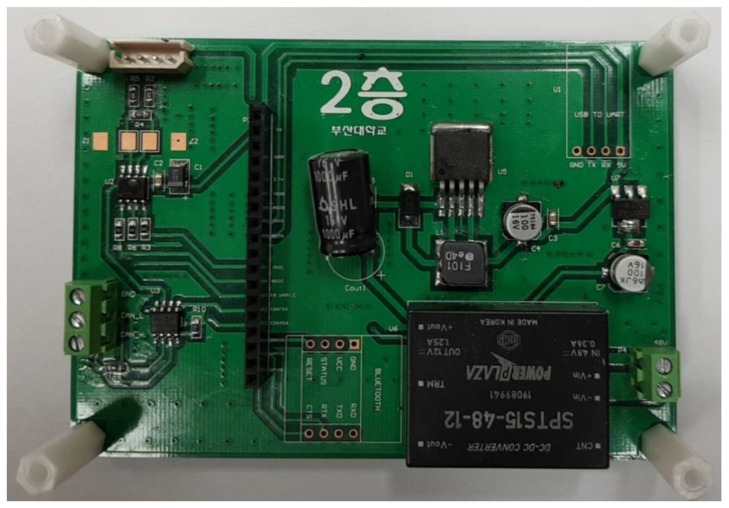
Bottom layer of Robot hand control module.

**Figure 22 sensors-20-06390-f022:**
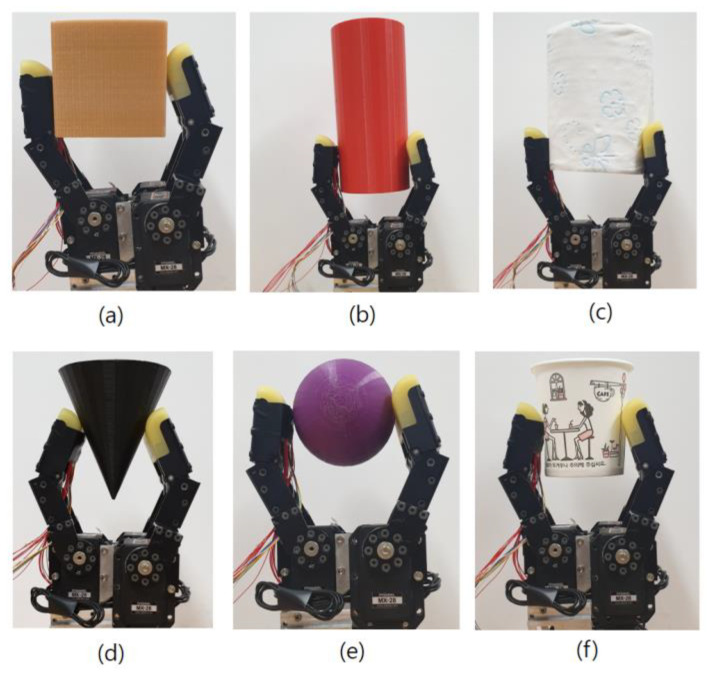
Grasping of various objects (**a**–**f**).

**Figure 23 sensors-20-06390-f023:**
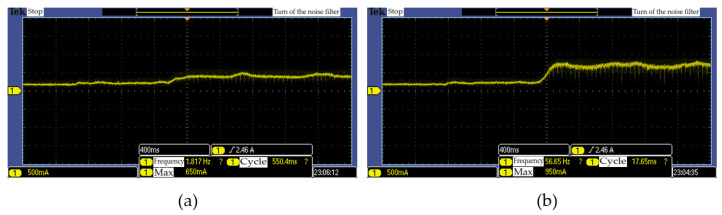
Current value of cylinder grasping: (**a**) torque min, (**b**) torque max.

**Figure 24 sensors-20-06390-f024:**
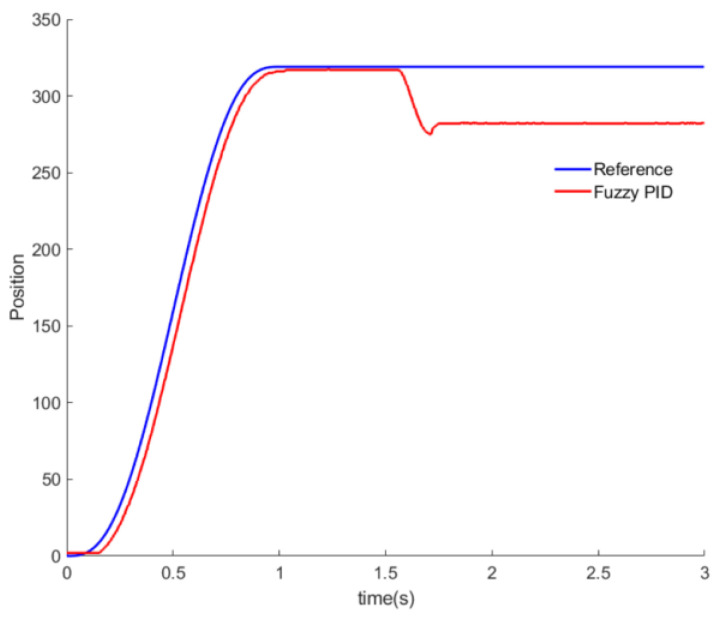
Grasp angle of the fuzzy proportional-integral-derivative (PID) controller.

**Table 1 sensors-20-06390-t001:** Weight expectation by deep neural network (DNN).

Object Weight (gf)	Sensor 1 (hpa)	Sensor 2 (hpa)	Sensor 3 (hpa)	Expect. Weight (gf)
0	1191.25	1199.02	1018.04	0.18
50	1191.69	1199.16	1018.38	50.14
100	1191.96	1199.55	1018.67	100.17
110	1192.02	1199.62	1018.71	110.12
120	1192.05	1199.65	1018.76	118.83
130	1192.07	1199.68	1018.79	130.27
140	1192.12	1199.74	1018.84	139.94
150	1192.14	1199.78	1018.86	150.19
160	1192.23	1199.83	1018.92	160.32
170	1192.25	1199.86	1018.96	170.18
180	1192.36	1200.01	1019.16	180.24
190	1192.41	1200.04	1019.17	189.81
200	1192.47	1200.06	1019.18	200.24

**Table 2 sensors-20-06390-t002:** Training data for the average the contact area.

Contact Force (kgf)	Sensor 1 (hpa)	Sensor 2 (hpa)	Sensor 3 (hpa)	Size	Average
0	1010.22	1010.12	1010.6	0	0
5	1017.52	1010.14	1026.75	1	1.1
5	1017.57	1010.15	1026.71	1	1.1
7	1028.35	1010.02	1054.76	1	1.1
7	1028.36	1010.04	1054.83	1	1.1
9	1045.37	1011.43	1071.78	1	1.1
9	1045.42	1011.42	1071.73	1	1.1
0	1010.35	1010.04	1010.5	0	0
5	1029.58	1011.16	1038.54	1	1.5
5	1029.57	1011.15	1038.66	1	1.5
7	1053.44	1010.73	1053.96	1	1.5
7	1053.35	1010.82	1053.94	1	1.5
9	1080.16	1013.27	1053.78	1	1.5
9	1080.2	1013.34	1053.76	1	1.5

**Table 3 sensors-20-06390-t003:** Fuzzy membership function.

Abbreviation	Meaning
NH	Negative Huge
NB	Negative Big
NM	Negative Medium
NS	Negative Small
ZO	Zero
PS	Positive Small
PM	Positive Medium
PB	Positive Big
PH	Positive Huge

**Table 4 sensors-20-06390-t004:** Specifications of grasping robot.

ID	Model	Spec
1 and 2	Maxon W10	DC motor 24 V, 150 W
3 and 4	Maxon W06	DC motor 24 V, 70 W
5	Maxon W01	DC motor 24 V, 20 W
Robot hand	Dynamixel (MX-28)	Coreless 12 V, RS485

**Table 5 sensors-20-06390-t005:** Min and max values of grasping torque.

Object	Torque Min	Torque Max
Cube	80	400
Cylinder	120	400
Cone	80	1023
Ellipsoid	50	400
Paper cup	50	1023
Scroll tissue	160	370

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
