# Peer review of "Artificial Intelligence-Based Optimal Grasping Control"

_sensors, 2020, doi:10.3390/s20216390_

Round 1
Reviewer 1 Report
This work describes the analysis of tactile sensors used for grasping in robotic applications using neural networks with the intent of using this as the basis of a more sophisticated manipulator control system in the future. The paper is well written with a reasonable amount of technical detail. There are many different types of tactile sensor and control systems in the literature, so the concepts here are not very revolutionary, but there is an increase in sophistication that will be appreciated by the readership.
A few specific comments
- The review of the literature is quite limited and there are many classes of tactile sensor not mentioned. It is worth expanding on this. In particular, there have been other tactile sensors for robots based on pressure sensors acted upon by fluids:
- Schmitz A, Maggiali M, Natale L, Bonino B, Metta G. A tactile sensor for the fingertips of the humanoid robot icub. In2010 IEEE/RSJ International Conference on Intelligent Robots and Systems 2010 Oct 18 (pp. 2212-2217). IEEE.
- Preechayasomboon P, Rombokas E. ConTact Sensors: A Tactile Sensor Readily Integrable into Soft Robots. In2019 2nd IEEE International Conference on Soft Robotics (RoboSoft) 2019 Apr 14 (pp. 605-610). IEEE.
- These, and related articles, should be reviewed and included.
- Similarly, the use of neural networks and fuzzy logic in this application is becoming more common. Review the field and emphasise why your work is novel in this context, the following may be useful:
- Yuan W, Mo Y, Wang S, Adelson EH. Active clothing material perception using tactile sensing and deep learning. In2018 IEEE International Conference on Robotics and Automation (ICRA) 2018 May 21 (pp. 1-8). IEEE.
- Pastor F, Gandarias JM, García-Cerezo AJ, Gómez-de-Gabriel JM. Using 3D Convolutional Neural Networks for Tactile Object Recognition with Robotic Palpation. Sensors. 2019 Jan;19(24):5356.
- Meier M, Patzelt F, Haschke R, Ritter HJ. Tactile convolutional networks for online slip and rotation detection. InInternational Conference on Artificial Neural Networks 2016 Sep 6 (pp. 12-19). Springer, Cham.
- Chen P, Hasegawa Y, Yamashita M. Grasping control of robot hand using fuzzy neural network. InInternational Symposium on Neural Networks 2006 May 28 (pp. 1178-1187). Springer, Berlin, Heidelberg.
- A couple more labels would be useful in Fig. 1 as it is not clear what all the parts are.
- Labels in Figure 2 - shouldn't these read silicone, not silicon? Also there is a space missing.
- Please explain, or add a reference, to explain 'Adam' on line 99
Author Response
Thank you for the kind reviewing.
The revised part of the paper is marked in blue.
Below is a description of the modified parts.
The references suggested by the reviewer were thoroughly reviewed and added to the article.
For the added references, [10,11] was added in line 37 in the introduction,
Comments for [16,17] and [18] have been added from lines 47 to 51.
As suggested by the reviewer, the label was added in figure 1 to clarify.
Also, to explain Adam, the reference paper [20] has been added to line 108.
Thanks to your review, it could be a better article.

Reviewer 2 Report
This paper in general can be greatly improved. The authors should be confident about the work and should be able to present more details. Some examples include:
- The basic principle of the sensor even though this part has been already published by the authors before. However, as a journal paper, the paper should be self-contented.
- Is it really necessary to use NN to train the model? I have a feeling some other machine learning techniques may be more effective for this problem such as SVM, GP.
- The literature review is definitely not sufficient and many related work in this domain should be considered. A google scholar search of "tactile sensing & grasp" will give many relevant works.
Author Response
Thank you for the kind reviewing.
The revised part of the paper is marked in blue.
Below is a description of the modified parts.
The references were thoroughly reviewed and added to the article.
For the added references, [10,11] was added inline 37 in the introduction,
Comments for [16,17] and [18] have been added from lines 47 to 51.
The model takes various inputs and predicts the contact area.
Therefore, we reviewed the models used for only one predicted value, and MLP was suitable for use.
The resulting graph for training and a description of it has been added to line 191.
Thanks to your review, it could be a better article.
